# HOW MUCH CAN WE FORGET ABOUT DATA CONTAMINATION?

## ABSTRACT

The leakage of benchmark data into the training data has emerged as a significant challenge for evaluating the capabilities of large language models (LLMs). In this work, we use experimental evidence and theoretical estimates to challenge the common assumption that small-scale contamination renders benchmark evaluations invalid. First, we experimentally quantify the magnitude of benchmark overfitting based on scaling along three dimensions: The number of model parameters (up to 1.6B), the number of times an example is seen (up to 144), and the number of training tokens (up to 40B). We find that if model and data follow the Chinchilla scaling laws, minor contamination indeed leads to overfitting. At the same time, even 144 times of contamination can be forgotten if the training data is scaled beyond five times Chinchilla, a regime characteristic of many modern LLMs. We then derive a simple theory of example forgetting via cumulative weight decay. It allows us to bound the number of gradient steps required to forget past data for any training run where we know the hyperparameters of AdamW. This indicates that many LLMs, including Llama 3, have forgotten the data seen at the beginning of training. Experimentally, we demonstrate that forgetting occurs faster than what is predicted by our bounds. Taken together, our results suggest that moderate amounts of contamination can be forgotten at the end of realistically scaled training runs.

## 1    INTRODUCTION

A core principle of machine learning is that a model should not be trained on the test set used for evaluation (Donoho, 2017). For foundation models trained on Internet-scale data, there are increasing concerns that this principle is violated due to the leakage of benchmark evaluation data into the training data (Xu et al., 2024; Oren et al., 2024). Indeed, many LLM developers have found overlap between their training data and the benchmark questions used for evaluation (Brown et al., 2020; Dubey et al., 2024). What is more, research on memorization (Carlini et al., 2019; 2021) shows that text sequences from the training data are sometimes encoded within the model, including machine learning datasets (Grynbaum & Mac, 2023; Liang et al., 2023; Nasr et al., 2023; Bordt et al., 2024).

While the fact that data contamination *can* lead to invalid performance evaluations is now well-established (Magar & Schwartz, 2022; Li & Flanigan, 2024; Yang et al., 2023; Jiang et al., 2024), little is known about the precise conditions under which this is the case. The main reason for this is that the training data is usually unknown, and contamination identified via clever research designs that work around this restriction (Golchin & Surdeanu, 2023; Oren et al., 2024; Deng et al., 2024). Because modern foundation models are sometimes trained for over a million gradient steps (Dubey et al., 2024), it is unclear whether a single update on contaminated data at some point during training necessarily impacts downstream evaluations. And indeed, there is quite some evidence that language models need to see samples repeatedly to have any impact on the final model. For example, many papers on memorization have found that it occurs only when a sample is frequently repeated in the training data (Carlini et al., 2022; Biderman et al., 2023; Huang et al., 2024b). The same is true for research on knowledge acquisition, where a fact needs to be paraphrased many times before it is finally remembered by the model (Allen-Zhu & Li, 2023; Cao et al., 2024; Chang et al., 2024).

In this work, we study the impact of data contamination in a controlled setting. This means we train language models from scratch on datasets where we explicitly insert contaminated examples (Jiang et al., 2024). We begin by quantifying how the overall magnitude of benchmark overfitting (or the

cross-entropy loss of an observed sample) changes as we **scale along three critical dimensions**: (1) the number of model parameters, (2) the number of training tokens, and (3) the number of repetitions of an example in the training data (Section 4.1). Holding the other two dimensions fixed, we find that *the effect of scaling is monotone in each dimension*. First, similar to many other works, we find that the tendency of a model to overfit increases in the number of parameters (Goodfellow et al., 2016; Zhang et al., 2017; Carlini et al., 2022). Second, and this is also expected, we find a clear scaling in the number of repetitions, where more frequently repeated observations exhibit stronger overfitting (Carlini et al., 2022; Huang et al., 2024b). More surprisingly, we find that the effect of contamination can *vanish* as we increase the number of training tokens, up to the point where 12 repetitions of an entire dataset in the training data have no impact on the downstream evaluation *on that same dataset*.

Our investigation reveals that the **natural forgetting** dynamics of gradient descent (Tirumala et al., 2022; Jagielski et al., 2023) is the reason why increasing the number of tokens alleviates the impact of contamination. Concretely, we show that training on five times Chinchilla (Hoffmann et al., 2022) of clean data can cause a model to forget even 144 times repeated training examples (Section 4.2). Forgetting the bulk of the impact of a training example can occur rapidly, a point that we demonstrate using OLMo-1B (Groeneveld et al., 2024) (Section 4.3). What is the reason for this rapid forgetting? We show that exposure to novel data is important. Interestingly, models tend to exhibit the strongest overfitting on examples seen repeatedly throughout training, even compared to those seen during the end (Section 4.2).

Because running pre-training experiments is expensive, we also ask to what degree forgetting can be explained by the **training dynamics of gradient descent**. We show that the weight decay parameter and learning rate schedule of the AdamW optimizer (Loshchilov & Hutter, 2019) play a key part in forgetting past training examples (Section 5). Concretely, we derive a simple theory of forgetting via cumulative weight decay (Section 5.1) and show that it provides an upper bound on empirical forgetting, which usually occurs faster (Section 5.2). The key point is that this approach allows us to gauge the degree of forgetting present in any training run for which the optimization hyperparameters are known. It even allows us to approximate how the final model parameters balance the gradient updates from different stages of training. Overall, our analysis indicates that many LLMs, including OLMo-7B and Llama 3 405B (Dubey et al., 2024), have forgotten the data seen at the beginning of their training run.

Taken together, our **main contribution** is to show that the impact of individual examples in the training data depends on the precise characteristics of the setting. There are settings where the effect can be significant; Chinchilla training is an important example (Section 4.1). However, there are equally realistic settings where individual examples don't matter - including quite likely the data-intensive training runs of many recent LLMs (Gemma Team, 2024).

## 2 RELATED WORK

**Data Contamination.** The GPT-3 paper (Brown et al., 2020) uses an n-gram-based approach to differentiate between "clean" and "dirty" benchmark questions. This approach has since been used in many LLM reports (Chowdhery et al., 2023; Touvron et al., 2023), including Llama 3 (Dubey et al., 2024), where it is estimated that there might be a performance gain of up to 8 and 14 percentage points on PiQA and HellaSwag, respectively. The GPT-4 technical report (Achiam et al., 2023) remarkably concluded that *"contamination overall has very little effect on the reported results"*. This has since given rise to a literature that aims to *detect* (Oren et al., 2024), *mitigate* (Li et al., 2024), and *estimate the effect of* (Yang et al., 2023; Bordt et al., 2024) data contamination under various assumptions, but crucially without access to the training data. This literature often challenges the conclusion that contamination overall has little effect in GPT-4 (Xu et al., 2024).

**Forgetting.** In machine learning, the term *forgetting* is frequently associated with *"catastrophic"* forgetting, where learning new tasks hurt the performance at previously solved tasks (Lopez-Paz & Ranzato, 2017). In the context of LLMs, catastrophic forgetting can occur during fine-tuning (Luo et al., 2023) or continual learning (Huang et al., 2024a). In contrast, this paper studies forgetting as a potential *"natural"* phenomenon of learning (Toneva et al., 2019). Tirumala et al. (2022) study forgetting in language modeling and find, similar to Toneva et al. (2019), that forgetting can be exponentially slow. In contrast, Jagielski et al. (2023) find that models empirically do forget examples

over time. In concurrent work, Pagliardini et al. (2024) propose to add a second momentum term to the AdamW optimizer, and show that this slows down the forgetting of past gradients.

**Data Attribution.** Data attribution methods (Koh & Liang, 2017; Ilyas et al., 2022; Park et al., 2023) aim to identify data points responsible for specific model behaviors. We ask how much a model's benchmark performance is influenced by seeing the example during training, which broadly falls within this field (Grosse et al., 2023; Choe et al., 2024). Importantly, we directly measure the influence of contaminated examples through retraining, avoiding the approximation errors that can occur when using data attribution methods (Koh & Liang, 2017; Ghorbani & Zou, 2019) for large-scale models (Basu et al., 2021; Bae et al., 2022)

## 3 BACKGROUND AND METHODS

This Section gives additional details on the research questions and lays out our experimental setup.

> **Research question**
>
> How does the presence of a text in the training data influence the final model's performance *on that same text*?

While it is well known that an individual data point can be influential if the training data and model are small (Koh & Liang, 2017), we are concerned with the large-data regime where the influence of any individual example may vanish (Basu et al., 2021; Bae et al., 2022). To further clarify this setup, Section 3.1 gives a brief overview of recent developments in scaling the training data of LLMs. Section 3.2 details the used benchmarks and explains how we contaminate the training data. Section 3.3 discusses the problem of near-duplicate benchmark questions.

**Models and Training Data.** We train language models of up to 1.6B parameters using the architecture and hyperparameters from the GPT-3 paper (Brown et al., 2020, Table 2.1). For this, we adopt the llm.c codebase. The training data is the 100BT split of the FineWeb-Edu dataset (Lozhkov et al., 2024). We also train OLMo-1B (Groeneveld et al., 2024) using the corresponding code and data (Soldaini et al., 2024).

### 3.1 WE CONSIDER THE REGIME OF N-TIMES CHINCHILLA

According to the Chinchilla scaling law, for every doubling of model size, the number of training tokens should also be doubled. The Chinchilla model itself has 70 billion (B) parameters and was trained on 1.4 trillion (T) tokens; suggesting that the number of training tokens should be roughly 20x the number of model parameters (Hoffmann et al., 2022, Table 3). While the Chinchilla paper was highly influential, modern language models are trained on significantly more tokens (Sardana & Frankle, 2024). For example, the OLMo-7B model was trained on 2.46T tokens, 17.5x the amount suggested by Chinchilla (Groeneveld et al., 2024). Similarly, the Llama 3 70B model was reportedly trained on 15T tokens, at over 10x Chinchilla (Dubey et al., 2024; Meta AI, 2024). The same holds for almost all recent LLMs at the 7B parameter scale (Gemma Team, 2024). In this paper, *we count the number of tokens a model is trained on as a multiple of its Chinchilla tokens*.

### 3.2 WE EVALUATE ON A MIX OF SEVEN DIFFERENT BENCHMARKS

We evaluate the impact of data contamination using a *mix* of seven different benchmarks: ARC-Easy (Clark et al., 2018), Social-I-QA (Sap et al., 2019), WinoGrande (Sakaguchi et al., 2021), PiQA (Bisk et al., 2020), BoolQ (Clark et al., 2019), MMLU (Hendrycks et al., 2021), and HellaSwag (Zellers et al., 2019). This means that every evaluation contains questions from all seven benchmarks. To construct the mixed contamination data, we first concatenate the different benchmarks. We then partition the set of all benchmark questions into subsets ranging from 10,000 to 2,000 questions so that each subset contains all benchmarks in equal weight: HellaSwag: 19.58%, SocialIQA: 8.27%, PiQA: 19.7%, MMLU: 21.82%, BoolQ: 6.48%, ARC-Easy: 5.92%, and WinoGrande: 18.16%. A holdout set of 10,000 benchmark questions is never added to the training data. The other subsets are

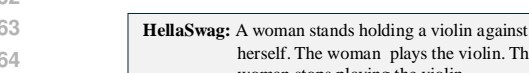

Figure 1: Language modeling benchmarks frequently contain near-duplicate questions. We perform extensive filtering for duplicates using fuzzy string matching. The figure depicts a near-duplicate from HellaSwag and a cross-benchmark duplicate from ARC-Easy/MMLU.

added to the training data, repeated either 4, 12, 36, or 144 times.[1] We consider *exact* contamination, that is we contaminate the training data with the same texts that the model is later evaluated on. We insert benchmark questions *individually* and at *random* positions into the training data. Models are evaluated zero-shot via the likelihood assigned to different sentence completions (Gao, 2021). For more discussion and details about how contamination is performed, see Supplement A.1 and Supplement A.2.

### 3.3 WE FILTER NEAR-DUPLICATE BENCHMARK QUESTIONS

Our method requires that there are no side effects from contaminating the training data with one question on the evaluation of another question. However, upon closer inspection, it turns out that *all* the commonly used benchmarks from the literature contain questions that are either near-duplicates or where the context of one question contains the answer to another question (for example, because the same text document was used to create multiple questions). This is illustrated in Figure 1, which depicts two near-duplicate questions on HellaSwag and questions from ARC-Easy and MMLU that are cross-benchmark duplicates. To address this problem, we perform extensive filtering, removing duplicate questions where the length-normalized Levenshtein distance falls below a certain threshold (Levenshtein, 1966; Navarro, 2001). This is documented in Supplement A.3, where we also describe an experiment to verify that our method is not invalidated by near-duplicate questions.

## 4 EXPERIMENTAL RESULTS

We now present our main experimental results. We being in Section 4.1 by discussing the scaling in model parameters, training tokens, and repetitions in the training data. The following Section 4.2 discusses various experiments on forgetting. The first two sections rely on training small GPT-3 models. Section 4.3 complements this with an analysis of OLMo-1B (Groeneveld et al., 2024).

### 4.1 CONTAMINATION SCALES WITH MODEL, DATA, AND REPETITIONS

We conduct three different experiments to understand how the effect of data contamination scales with the number of model parameters, training tokens, and the number of times a contaminated example is seen. First, we train increasingly large models on the same dataset of 7B tokens. Second, we train 124M parameter models on increasingly many tokens. Third, we train increasingly large models according to the Chinchilla scaling laws (Hoffmann et al., 2022), meaning that the number of training tokens scales linearly with the model parameters. In all experiments, we contaminate the training data *uniformly at random* with benchmark questions.

Figure 2 depicts the results of all three experiments. Because we are interested in the performance *difference* between the holdout data and the contaminated examples, Figure 2 depicts the *accuracy gap* between the holdout and contaminated examples in percentage points. In Figure 2a, we see that the accuracy gap due to contamination is *increasing in the number of model parameters*. For a 124M

---

[1]In preliminary experiments, we found that these numbers pleasantly cover the range from statistically significant contamination to complete overfitting. We also considered repeating observations a single time, as in (Jiang et al., 2024). However, we found this often leads to accuracy differences of about one or two percentage points, just within the margin of our confidence intervals, which is undesirable.

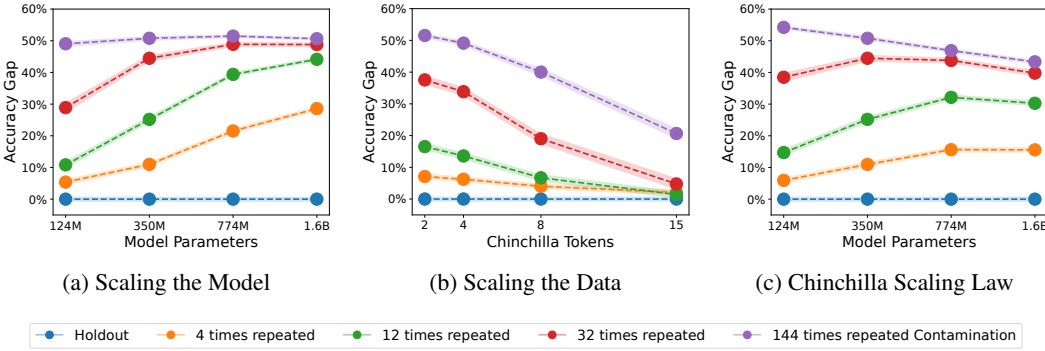

(a) Scaling the Model    (b) Scaling the Data    (c) Chinchilla Scaling Law

Figure 2: **Benchmark overfitting due to contamination.** **(a)** We train different models on 7B tokens. **(b)** We train 124M parameter models on increasingly many tokens. **(c)** We train models according to the Chinchilla scaling laws. The figure depicts the accuracy difference in percentage points between the holdout (normalized to zero) and the contaminated examples. The results are across a mix of seven different benchmarks, as outlined in Section 3.2. Different colors indicate different levels of contamination. Mean and bootstrapped 90% confidence intervals.

parameter model trained on 7B tokens, the overfitting due to 4 times contamination is 5 percentage points. For a 1.6B parameter model train on the same dataset, it is 20. Next, Figure 2b shows that the accuracy gap is *decreasing in the number of training tokens*. For a 124M parameter model trained at 2x Chinchilla, the accuracy gap due to 12 times contamination is 18 percentage points. For a 124M parameter model trained at 15x Chinchilla, the same accuracy gap is within the confidence interval of the holdout. From Figure 2, we also see that the accuracy gap is *increasing in the number of times an example is repeated*. For a 350M parameter model trained on 7B tokens, the accuracy gap is 11, 25, 44, and 51 percentage points for 4, 12, 32, and 144 times repeated contamination, respectively.

Because the accuracy gap *increases* in the number of model parameters and *decreases* in the number of tokens, the interesting question is how it behaves if model parameters and tokens are scaled *jointly*. A natural starting point is to double the number of training tokens for every doubling of model parameters, as specified by the Chinchilla scaling laws (Hoffmann et al., 2022). Figure 2c depicts the accuracy gap due to

Table 1: Accuracy of the Chinchilla models.

| Model | Holdout | 4x | 12x | 32x | 144x |
|-------|---------|-------|-------|-------|-------|
| 124M  | 42.22   | 48.14 | 56.92 | 80.70 | 96.45 |
| 350M  | 44.72   | 55.69 | 69.90 | 89.20 | 95.50 |
| 774M  | 49.16   | 64.76 | 81.30 | 92.95 | 96.05 |
| 1.6B  | 52.06   | 67.61 | 82.32 | 91.85 | 95.40 |

contamination as we train increasingly large Chinchilla-optimal models. While there is no clear monotone pattern, we see that moderate amounts of contamination can lead to significant overfitting. For the 774M parameter model, 4 times repeated contamination leads to an accuracy gap of 15 percentage points, suggesting that *under Chinchilla training, a single time of contamination can lead to overfitting of as much as 3 percentage points.*[2]

## 4.2 Contamination can be completely Forgotten

In the previous Section 4.1, we saw that the accuracy gap due to contamination decreases in the number of tokens up to the point where even 12 repetitions of a benchmark question in the training data can become insignificant. In this Section, we identify the natural forgetting dynamic of neural network training as the reason for this effect. We discuss how quickly forgetting occurs, whether examples are completely forgotten, and what kind of repetition makes a model remember.

To study the effect of forgetting, we train a 124M parameter model for 15 epochs. Instead of contaminating uniformly over the course of training like in the previous Section 4.1, we perform

---

[2]To contaminate the training data of a 774M model a single time with 10,000 benchmark questions, we need to insert ~0.5 million tokens into 15.5 billion training tokens, about 0.003% of the training data.

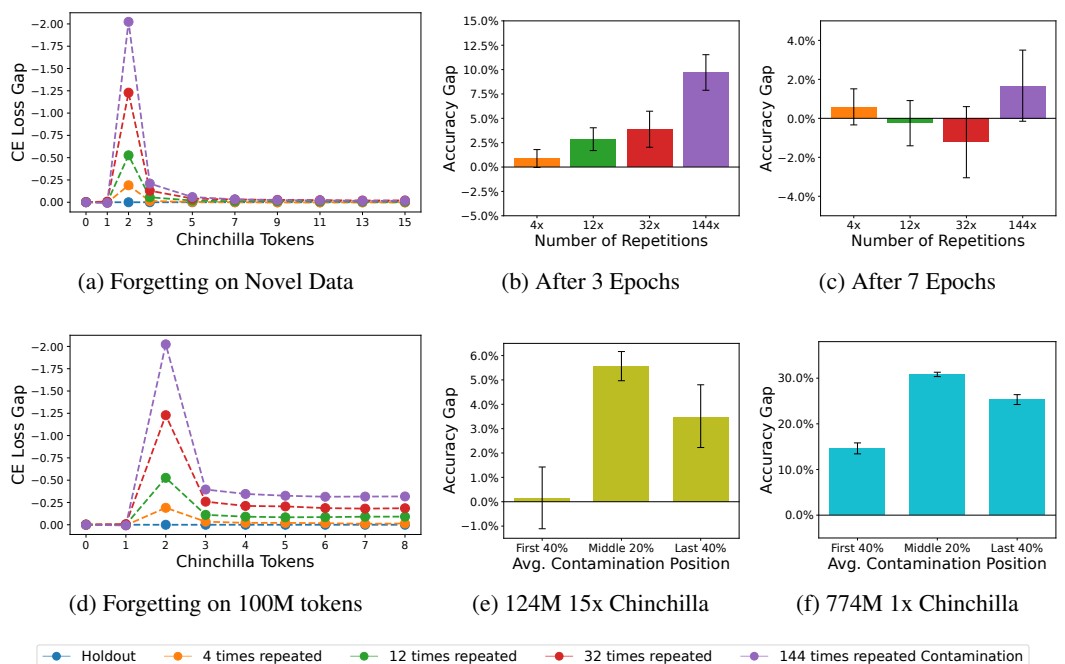

Figure 3: **The natural forgetting dynamic of neural network training. (a)** The development of the cross-entropy loss difference between contaminated and holdout benchmark questions over the course of training. Contamination occurs between the first and second Chinchilla (1 and 2 on the x-axis). **(b)** Accuracy gaps after training for 3 Chinchilla. **(c)** Accuracy gaps after training for 7 Chinchilla. **(d)** Same as (a). **(e)+(f)** The accuracy gap depends on the average position of an example in the training data. Mean and bootstrapped 90% confidence intervals.

the contamination between the first and second Chinchilla.[3] Figure 3a depicts the development of the *difference* in cross-entropy loss between contaminated and clean benchmark questions over the course of training. We see a strong peak after 2 Chinchilla, which is expected and shows the effect of contamination. What is interesting to us is the rate at which the cross-entropy loss difference decays as we continue training. After training for 1 additional Chinchilla (2.5B tokens for the 124M parameter model), it has already decayed significantly. However, the difference is still visible in Figure 3a. Figure 3b depicts the corresponding accuracy gaps at this point, and we see that all contamination levels still lead to overfitting. As we continue training, the cross-entropy loss difference between contaminated and holdout questions further narrows. From Figure 3c, which depicts the accuracy gaps after forgetting for a total of 5 Chinchilla, we see that the effect of contamination is eventually *completely forgotten* in the sense that there is no longer any accuracy difference between contamination and holdout benchmark questions.

The result that contamination can be completely forgotten is in contrast to some previous work on forgetting which have found that forgetting approaches a stable baseline Tirumala et al. (2022, Figure 10), or that certain examples are never forgotten (Toneva et al., 2019). To understand this difference, observe that many previous works on forgetting have not trained on a continuous stream of data. Instead, they have trained on the same training set for multiple epochs. Consequently, we modify our forgetting experiment to repeatedly train on the same 100M tokens after the second epoch. The result of this experiment is depicted in Figure 3d and should be compared to Figure 3a. Interestingly, this simple modification causes the effect of forgetting to stabilize at a level strictly larger than zero. We conclude that *exposure to novel data is important for forgetting*, an observation similar to Jagielski et al. (2023).

---

[3]Note that the model is already fairly trained after the first Chinchilla, meaning that the contamination is not very early during training. This is important because there is evidence that observations are more quickly forgotten if the model has not yet learned representations (Jagielski et al., 2023; Cao et al., 2024; Huang et al., 2024b). This is *not* the setting we are studying here.

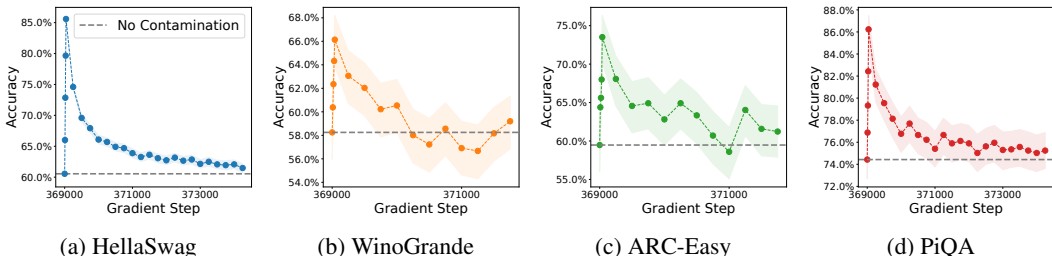

(a) HellaSwag      (b) WinoGrande      (c) ARC-Easy      (d) PiQA

Figure 4: **Contamination and forgetting in OLMo-1B.** We contaminate the OLMo-1B checkpoint at gradient step 369,000 four times with different benchmarks. This causes an average accuracy increase of 15 percentage points. We then continue pre-training for 1% of the remaining training time, leading to a reduction of 96% of the accuracy increase due to contamination. In this figure, different colors simply correspond to different benchmarks, and the grey line depicts the clean accuracy without contamination. Mean and bootstrapped 90% confidence intervals.

To further understand the impact of forgetting, we now ask whether examples seen late during training influence model behavior more strongly than examples seen early during training. To study this question, we average all the different *uniform* contamination levels from the models in the previous Section 4.1 (to gain statistical power) and consider the amount of overfitting depending on whether a question is seen, on average, in the beginning, middle, or end of training. The result of this experiment is depicted in Figure 3e and Figure 3f. As expected under forgetting, we see that benchmark questions seen early during training exhibit the smallest amount of overfitting. Interestingly and somewhat unexpectedly, questions that are neither clustered towards the beginning nor the end but as uniformly distributed throughout training as possible exhibit the strongest overfitting, suggesting that this spaced form of repetition helps the model remember (the middle peak is the most pronounced both in Figure 3e and Figure 3f).

### 4.3 Contamination and Rapid Forgetting in OLMo-1B

In the previous sections, we trained small GPT-3 models from scratch. In this Section, we complement this analysis by pre-training from an intermediate OLMo-1B checkpoint (Groeneveld et al., 2024). Similar to the analysis in Section 4.2, we insert the benchmark data at a specific point into the training data and then measure the subsequent forgetting. Unlike in the previous Section, we now insert the entire benchmark data – we already have a "clean" baseline from the original OLMo-1B training run. We insert each benchmark question four times and contaminate with four different benchmarks: HellaSwag (Zellers et al., 2019), WinoGrande Sakaguchi et al., 2021, ARC-Easy (Clark et al., 2018), and PiQA (Bisk et al., 2020).

Figure 4 depicts the result of the experiment. The effect of contamination is visible from the five leftmost points of every plot. The leftmost point corresponds to the uncontaminated model, and the next four points each depict the effect of one time contamination. Again, we see that *the immediate effect of contamination is significant*, leading to an average accuracy increase of 15 percentage points across the different benchmarks. At the same time, we also see that *the effect of contamination decays considerably as we continue training*. To contextualize this result, note that Figure 4 depicts less than 2000 gradient steps. The pre-training stage of OLMo-1B model consists of 739,328 gradient steps. This means that Figure 4 depicts less than 1% of the total forgetting until pre-training is done.

## 5 What is the role of weight decay in forgetting?

In the previous Section 4, we have seen that forgetting is an important empirical property of LLM training. In this section, we show that the weight decay parameter and learning rate schedule of the AdamW optimizer play a key part in forgetting past training examples. This offers a novel perspective on the interplay between these two parameters, usually seen in terms of generalization and training stability (Van Laarhoven, 2017; Zhang et al., 2019; Lewkowycz & Gur-Ari, 2020; Andriushchenko et al., 2023).

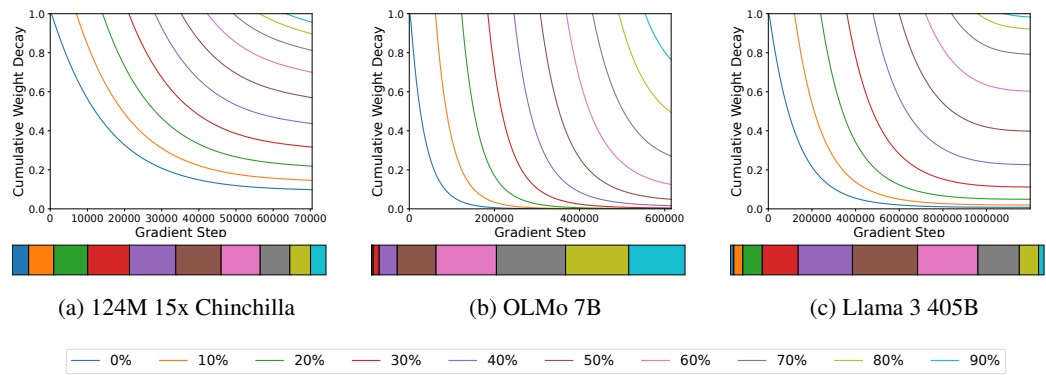

(a) 124M 15x Chinchilla  (b) OLMo 7B  (c) Llama 3 405B

— 0%  — 10%  — 20%  — 30%  — 40%  — 50%  — 60%  — 70%  — 80%  — 90%

Figure 5: **Theoretical estimates of forgetting and approximate model weight composition at the end of training.** *Top Row:* The cumulative weight decay $w_{t_1}^{t_2}$ as defined in equation (3) for different training runs. The figures depict the decay of the gradient updates for every decile of the training run, indicated in different colors. *Bottom Row:* The approximate composition of the final model weights in terms of the gradient updates from different deciles of the training run. The deciles are indicated in colors depicted in the legend below the plot.

### 5.1 WEIGHT DECAY AS A MECHANISM FOR FORGETTING TRAINING EXAMPLES

Consider the parameter update of AdamW at gradient step $t \geq 1$. It consists of two decoupled updates (Paszke et al., 2019; PyTorch Contributors, 2024): A weight decay update given by

$$\hat{\theta}_t = \theta_{t-1} - \gamma \lambda_t \theta_{t-1}, \tag{1}$$

and a gradient update given by

$$\theta_t = \hat{\theta}_t - \lambda_t \, \hat{m}_t / (\sqrt{\hat{v}_t + \epsilon}). \tag{2}$$

Here, $\theta_t$ are the model parameters, $\lambda_t$ is the learning rate, $\gamma$ is the weight decay parameter, and $\hat{m}_t$ and $\hat{v}_t$ are first- and second-order moment estimates of the gradient. Denoting the model weights at initialization by $\theta_0$, and the adaptive gradient by $\hat{g}_t = \hat{m}_t / (\sqrt{\hat{v}_t + \epsilon})$, we can iterate (1) and (2) to obtain

$$\theta_T = w_0^T \theta_0 - \sum_{t=1}^{T} w_t^T \lambda_t \, \hat{g}_t \qquad \text{where} \quad w_{t_1}^{t_2} = \prod_{i=t_1+1}^{t_2} (1 - \lambda_i \gamma). \tag{3}$$

Here, the weights $w_{t_1}^{t_2}$ account for the *cumulative weight decay* between gradient step $t_1$ and $t_2$. Intuitively, the model weights after $t$ gradient steps are a weighted average of the initial model weights and all the adaptive gradient updates up to time step $t$. This is not specific to the AdamW optimizer and applies to every optimizer with weight decay. Analyzing equation (3) reveals a critical factor influencing forgetting: The exponential decay of the $w_{t_1}^{t_2}$ with respect to increasing gap $t_2 - t_1$ in the optimization steps. In other words, the longer an update occurs in the past (i.e., the larger $t_2 - t_1$), the more the contribution of the update $\hat{g}_{t_1}$ is being scaled down due to the exponential decay of weights $w_{t_1}^{t_2}$. We can describe the evolution of these weights as the function of the time $T = t_2 - t_1$.

**Proposition 1.** *(The Decay of Past Gradients) The number of optimization steps $T = t_2 - t_1$ that are required to make the contribution of a model update at time $t_1$ small, that is $w_{t_1}^{t_2} \leq \epsilon$ for some small $\epsilon \in \mathbb{R}^+$, scales as $T \gtrsim \frac{\log(1/\epsilon)}{\gamma \lambda_{avg}}$, where $\lambda_{avg} = \frac{1}{T} \sum_{t=t_1+1}^{t_2} \lambda_t$ is the average learning rate of the optimizer between $t_1$ and $t_2$.*

We present a proof in the Supplement A.5. If $w_{t_2}^{t_1} \leq \epsilon$ for some sufficiently small $\epsilon$, the term $w_{t_2}^{t_1} \lambda_{t_1} \hat{g}_{t_1}$ vanishes from the sum in (3). Intuitively, this is the same as saying that the gradient update at time step $t_1$ has been forgotten.

**We now analyze the weight-decay mechanism of forgetting in different LLM training runs.** Of course, the decay of past gradients described in Proposition 1 is only one effect that contributes to

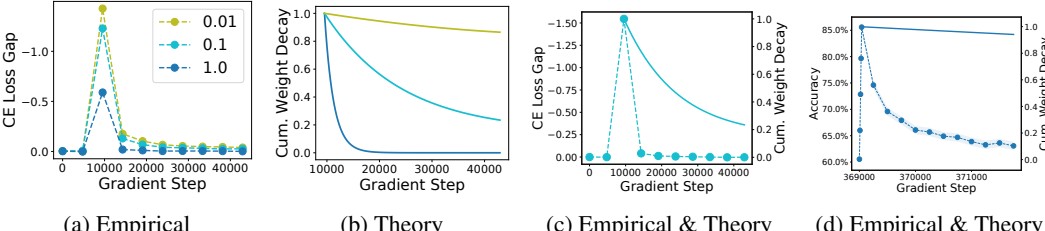

(a) Empirical        (b) Theory       (c) Empirical & Theory     (d) Empirical & Theory

Figure 6: **The theoretical estimates provide an upper bound on empirical forgetting, which can occur much faster. (a)** Empirical forgetting for three different weight decay parameters. **(b)** The corresponding cumulative weight decay. **(c)** Empirical forgetting and theoretical estimate for the default weight decay of 0.1. The y-axis of the cumulative weight decay $w_{t_1}^{t_2}$ is calibrated to start at the peak of the empirically observed overfitting. **(d)** Empirical forgetting and theoretical estimate for OLMo-1B (HellaSwag).

forgetting. Indeed, there is also the potential for different gradient updates in the sum (3) to cancel each other. However, because a term decayed to zero is definitively forgotten, we can consider the cumulative weight decay as an *upper bound* on forgetting. Figure 5 depicts the evolution of the forgetting term $w_{t_1}^{t_2}$ for the 124M parameter model from Section 4.2, OLMo 7B, and Llama 3 405B (where we assume the model trained with a weight decay of 0.1). For the training data at each decline of the training run, Figure 5 depicts *forgetting curves* that indicate how much the corresponding gradients decay as we continue training. For the 124M model depicted in Figure 5a, even the initialization is not completely decayed towards the end (the blue curve is still strictly larger than zero). In contrast, for OLMo-7B, the gradients of the first 40% of the training data decay to zero until the end of training, meaning that this data is forgotten (Figure 5b). Llama 3 405B also experiences significant decay of the early gradients (Figure 5c).

**The interplay between the weight decay parameter and learning rate schedule of AdamW** creates an interesting dynamic. Lower learning rates towards the end of training (1) slow down the forgetting of past training examples and (2) decrease the impact of later training examples on the final model weights. To better understand this dynamic, we plot the sum of the terms $\lambda_{t_1} w_{t_1}^{t_2}$ for each decile of the training run, normalized by the same sum over the entire training run. This is depicted in the bottom row of Figure 5 and can be thought of as a simple approximation of how the final model weights are composed by the gradients of different training deciles. Interestingly, this approximation suggests that the Llama 3 405B training run, where supposedly a lot of expertise has gone into the choice of the hyperparameters, results in a model where the approximate influence of different training steps is symmetrically distributed around the middle of training (Figure 5c). In contrast, the OLMo-7B training could seemingly benefit from further decaying the learning rate to give more weight to early versus late gradients.

While our analysis in this Section considers the mechanistic effect of individually decaying gradient updates, it does not model any interactions between different gradient updates. For example, if the weight updates at a later time step $t_2$ were aligned with past updates at $t_1$, then the model might not forget the information even if the effect of past updates vanishes from the sum. However, such complex interactions are avoided if the model updates from contaminated samples are orthogonal. Formalizing this observation leads to a more rigorous version of Proposition 1, presented in Supplement A.5. The argument is that under suitable gradient orthogonality conditions, the decay of past gradients can guarantee forgetting.

## 5.2 PRACTICAL FORGETTING OCCURS FASTER THAN WHAT THE THEORY PREDICTS

In the previous Section 5.1, we have derived a simple theory of forgetting via cumulative weight decay. We now investigate how the theoretical estimates relate to the empirically observed forgetting.

The main parameter that controls the theoretical forgetting curves is the weight decay parameter. Therefore, we ask how forgetting changes empirically when we change the weight decay. Figure 6a depicts the result of repeating the forgetting experiment from Section 4.2 with three different choices for the weight decay parameter. From Figure 6a, we see that the weight decay parameter

controls the impact of contamination at all time steps, where a larger weight decay parameter leads to more forgetting and a smaller weight decay parameter to less forgetting. This is consistent with the theoretical predictions depicted in Figure 6b, meaning there is a *qualitative alignment between the empirical results and our theoretical predictions*.

To better understand the quantitative relation between empirical forgetting and the theoretical estimates, we ask how the empirically forgotten fraction (of cross-entropy loss or accuracy) relates to the cumulative weight decay. Figure 6c depicts the empirical decay and corresponding theoretical prediction for the model from Section 4.2. We see that the theoretical estimate is somewhat pessimistic and that *forgetting occurs faster than what is predicted by the theory*. Figure 6d is similar to Figure 6c, except that we consider the OLMo-1B forgetting experiment from Section 4.3. Figure 6d depicts a case where *forgetting occurs much faster* than what is predicted by the theory. Interestingly, we also see that the empirical rate of forgetting, at least in this experiment, is not smaller for the larger model.

## 6 DISCUSSION

This work presents different experiments on data contamination and forgetting. We have seen that the impact of contamination can vanish as the size of the training data increases – an aspect that has largely been overlooked in the literature (Yang et al., 2023; Oren et al., 2024; Jiang et al., 2024). We have also shown that the hyperparameters of AdamW play an important part in forgetting – an insight that might inform the parametrization of future training runs. Of course, it would be interesting to study contamination and forgetting on a larger scale. Nevertheless, we have shown that the effects of contamination can vanish *even though* the number of steps we could train the models for was limited.

We have studied data contamination with a focus on the leakage of benchmark questions into the training data. This means that our work might be more informative about the topic than other works that study contamination in different contexts. At the same time, one has to be careful when extrapolating our results, especially to a privacy setup (Carlini et al., 2019; Jagielski et al., 2023). This is because empirical forgetting might behave differently for random strings or otherwise uniquely identifiable information (Carlini et al., 2021).

The *"contamination problem"* as studied in this paper has interesting connections to many areas of machine learning, including privacy (Graves et al., 2021; Jagielski et al., 2023), data attribution (Kirchenbauer et al., 2024), and generalization (Bousquet & Elisseeff, 2002; Hardt et al., 2016; Mania et al., 2019). The connection to data attribution is especially relevant in light of our results. This is because we demonstrate cases where the presence or absence of a datapoint in the training data is irrelevant for the model behavior *on that same datapoint*, meaning it does not make sense to attribute model behavior to individual datapoints in this regime.

## 7 REPRODUCIBILITY STATEMENT

The results in this paper were obtained using the OLMo codebase, available at `https://github.com/allenai/OLMo`, and the llm.c codebase, available at `https://github.com/karpathy/llm.c`. Our code is fully reproducible, including the random positions at which benchmark questions were inserted into the training data. We trained on the 100BT split of the FineWeb-Edu dataset, available at `https://huggingface.co/datasets/HuggingFaceFW/fineweb-edu`. Anonymized code for this paper is available at `https://github.com/iclr10261/code`.

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

## A APPENDIX

### A.1 ADDITIONAL DISCUSSION OF DATA CONTAMINATION ASSUMPTIONS AND SETTING

Here, we discuss our data contamination approach in a bit more detail.

In this paper, we consider only **exact contamination**. This means we contaminate the training data exactly with the text the model is later evaluated on. In the literature, it has been shown that non-exact contamination (re-worded questions, translation into a different language) can affect benchmark performance, too. For example, Yang et al. (2023) have shown that a 13B parameter Llama 2 Model (Touvron et al., 2023) can achieve an accuracy increase of over 20 percentage points after training on re-phrased benchmark questions. We decided against considering non-exact contamination for this paper because the models we train from scratch are much smaller than those for which non-exact contamination results have been shown. This means these models are less capable of making sense of related information, potentially leading us to underestimate the effect of non-exact contamination for realistic training runs.

In addition, we consider contamination with **individual benchmark questions**, inserted into the training data at **random** positions. We consider this setup because we are interested in contamination from the perspective of *leakage*, where individual benchmark questions may enter the training data via different documents (for example, as quotes in Wikipedia articles, a case described in Brown et al. (2020)). This contrasts with the setup where a dataset is present in the training data as a long contiguous string, which we conjecture might have a similar impact but be easier detectable (Oren et al., 2024). The fact that we contaminate with benchmark questions also sets us apart from related works that study data contamination and memorization for random strings and uniquely identified objects (Carlini et al., 2019; 2021). It is worth highlighting that the results between these two setups might differ, especially considering the time it takes to forget an example.

We only consider pre-training.

### A.2 ADDITIONAL DETAILS ON EVALUATION AND HOW CONTAMINATION WAS PERFORMED

**Benchmark Questions and Evaluation.** We use code from OLMo (Groeneveld et al., 2024) to format the different benchmark questions. This code is again based in part on the EleutherAI Evaluation Harness (Gao et al., 2024). The benchmark questions are multiple-choice, and the different options are presented to the model zero-shot as possible sentence continuations. The prediction is the sentence continuation with the largest likelihood. For the small GPT-3 models, we normalize by the number of tokens (Gao, 2021). For OLMo, we rely on the evaluation framework that is part of the code repository.

**Inserting benchmark questions into the training data.** A batch of LLM training data consists of $B$ sequences of $S$ tokens, resulting in a batch size of $B \times S$. For example, OLMo-1B is trained with $B = 2048$ and $S = 2048$; the batch for a single gradient step contains ~4M tokens (Groeneveld et al., 2024). Individual sequences in a batch usually contain multiple texts separated by a special end-of-text token. We insert benchmark questions at random positions into the pre-training data, separated at the beginning and end with the end-of-text token.

### A.3 FILTERING NEAR-DUPLICATE BENCHMARK QUESTIONS

Upon close inspection of the different benchmarks, it turns out that there are exact and near-duplicate questions both within and across benchmarks. Consider, for example, the following two examples from HellaSwag (Zellers et al., 2019):

> **HellaSwag 1:** *A person is seen riding a board along the water with a kite on top. more clips are shown of the person riding back and fourth on the board.*

and

> **HellaSwag 2:** *A person is seen riding a board along the water with a kite on top. More clips are shown of the person riding back and fourth on the board. the person continues to ride the board along the water.*

Table 2: Overview of benchmarks used in the paper. This table documents the experiments with GPT-3 models. The first two rows provide the dataset split and corresponding number of benchmark questions. The third row provides the number of questions that were removed from the dataset after filtering each dataset for near-duplicate questions. The fourth row provides the number of questions that were removed after additionally filtering for near-duplicate questions across all the different datasets combined. The fifth row provides the dataset's weight in the dataset splits used in the experiments.

| | HellaSwag | PiQA | Social-i-QA | BoolQ | MMLU | WinoGrande | ARC-Easy |
|---|---|---|---|---|---|---|---|
| **Split** | Validation | Train | Train | Validation | Test | XL, Train | All |
| **Size** | 10,042 | 16,113 | 33,410 | 3,269 | 14,042 | 40,398 | 5,197 |
| **Filtered** | 1,416 | 7,386 | 29,756 | 409 | 4,423 | 21,944 | 2,568 |
| **Cross-Filtered** | 3 | 6 | 10 | 2 | 15 | 0 | 13 |
| **Weight** | 19.58% | 19.77% | 8.27% | 6.48% | 21.82% | 18.16% | 5.92% |

Note that these are the ground-truth options of two *different* benchmark questions. Similar patterns can be observed for many benchmarks, for example, because a single text document was used to create different benchmark questions. Here is another example from PiQA (Bisk et al., 2020):

> **PiQA 1:** *Goal: how do you close a cabinet? Solution: push the door shut.*

and

> **PiQA 2:** *Goal: how do you close a cabinet? Solution: shut the door.*

Near-duplicate benchmark questions present a challenge for our methodology. This is because one question might end up in a set of benchmark questions that we highly contaminate with, whereas the other question might end up in the purportedly "clean" set of benchmark questions that we use for holdout evaluation. To tackle this problem, we perform fuzzy string matching between the ground-truth options (that is, the potential contamination data) of all benchmark questions, randomly removing one question for every detected duplicate. We use the Python package rapidfuzz.

**Summary Statistics.** Table 2 depicts summary statistics about the different benchmarks, including the number of questions that were filtered during the duplicate-detection stage. We see that the number of filtered questions is significant. On some datasets, especially Social-i-QA, we had to apply very aggressive filtering to avoid any side-effects during contamination. Hence, the number of removed questions per dataset does not necessarily reflect the actual number of duplicates, but the level of filtering that had to be applied to remove all duplicate questions.

**Experimental verification that filtering worked.** We verify that our filtering procedure worked by training two models: One that is heavily contaminated (obtaining an accuracy of over 97%), and another model that did not see any contamination. We then evaluate both models on a set of 10,000 benchmark questions that are holdout even for the contaminated model. The contaminated model obtains an accuracy of 42.2%, (95%-CI: 41.2% - 43.2%) on the holdout, while the clean model obtains an accuracy of 41.9% (95%-CI: 41.0% -42.9%). Because the observed accuracy difference is small in absolute terms and lies within the confidence interval, we conclude that there are no significant side-effects in our evaluation procedure.

A.4   PROOF OF PROPOSITION 1

**Proposition 2.** *(The Decay of Past Gradients) The number of optimization steps $T = t_2 - t_1$ that are required to make the contribution of a model update at time $t_1$ small, that is $w_{t_1}^{t_2} \leq \epsilon$ for some small $\epsilon \in \mathbb{R}^+$, scales as $T \gtrsim \frac{\log(1/\epsilon)}{\gamma \lambda_{avg}}$, where $\lambda_{avg} = \frac{1}{T} \sum_{t=t_1}^{t_2} \lambda_t$ is the average learning rate of the optimizer between $t_1$ and $t_2$.*

*Proof.* Without loss of generality, mapping $t_1 = 1$ and $T = t_2$, we have from equation 3:

$$w_1^T = \prod_{i=1}^{T}(1 - \lambda_i \gamma)$$

Assigning the forgetting ratio to be less than $\epsilon$ according to our criteria, we have:

$$\prod_{i=1}^{T}(1 - \lambda_i \gamma) \leq \epsilon$$

$$\sum_{i=1}^{T} \log(1 - \lambda_i \gamma) \leq \log \epsilon$$

$$\sum_{i=1}^{T}(-\lambda_i \gamma) \lesssim \log \epsilon \quad (\log(1 - x) \approx -x \text{ for small } x)$$

$$T \times \left(\frac{1}{T}\sum_{i=1}^{T} \lambda_i\right) \times \gamma \gtrsim \log \frac{1}{\epsilon}$$

Re-arranging this equation gives us the desired result, where $\lambda_{avg} = (\frac{1}{T}\sum_i \lambda_i)$. $\qquad \square$

### A.5 Extended Analysis of Forgetting & Gradient Alignment

To understand the effect of weight decay on forgetting, we analyze two different stages of optimization: (1) the contamination stage, where the training set consists of only the contaminated samples, and (2) the forgetting stage, where the training set consists only of clean samples. Here, we consider the SGD learning algorithm, but the resulting analysis also applies to SGD with momentum. In particular, to illustrate the effect of weight decay, we assume the usage of SGD for the contamination stage and SGD with weight decay for the forgetting stage.

We now introduce some notation. Let $\theta \in \mathbb{R}^D$ be the weights of the model, and let $\mathcal{X}_{\text{cont}} = \{\mathbf{x}_1^{\text{cont}}, \mathbf{x}_2^{\text{cont}}, ...\mathbf{x}_{N_{\text{cont}}}^{\text{cont}}\}$ be the contamination set and $\mathcal{X}_{\text{clean}} = \{\mathbf{x}_1^{\text{clean}}, \mathbf{x}_2^{\text{clean}}, ...\mathbf{x}_{N_{\text{clean}}}^{\text{clean}}\}$ be the clean pre-training set. The training data used for the contamination stage is thus $\mathcal{X}_{\text{cont}}$, and the training data for the forgetting stage is $\mathcal{X}_{\text{clean}}$. Let $\ell(\mathbf{x}_i) \in \mathbb{R}$ be the loss associated with sample $\mathbf{x}_i$. Let the model be initialized at the contamination stage with $\theta = \theta_{\text{init}}$. The learning algorithm is (single batch size) SGD, which is run for a total of $N_{\text{cont}}$ steps for the contamination stage and $N_{\text{clean}}$ steps for the forgetting stage. First, we observe that the weights at the end of the contamination stage are:

$$\theta' = \theta^{\text{init}} - \underbrace{\sum_{i=1}^{N_{cont}} \lambda_i \nabla_{\theta i} \ell(\mathbf{x}_i^{\text{cont}})}_{\theta^{\text{cont}}}$$

We thus denote the weights at the end of the contamination stage as $\theta' = \theta^{\text{init}} + \theta^{\text{cont}}$. For the subsequent fine-tuning stage to forget information regarding samples $\mathcal{X}_{\text{cont}}$, we first define a criterion to identify forgetting based on the angle between a weight vector and the contaminated part identified above.

**Forgetting Criteria.** A model with weights $\theta$ is said to have "forgotten" information contained in $\theta^{\text{cont}}$ if $\frac{|\theta^\top \theta^{\text{cont}}|}{\|\theta^{\text{cont}}\|_2^2} \leq \epsilon$, which we call the "forgetting ratio". Here, $\epsilon \in \mathbb{R}^+$ is a small constant.

We now proceed with an analysis of the forgetting stage. To enable this, we make the following important assumption that the gradients of clean and contaminated samples are orthogonal for all clean and contaminated samples across all optimization steps. This is a relatively strong assumption, and it quantifies the intuition that model updates required by SGD to memorize clean samples and contaminated samples are distinct.

**Assumption 1. (Gradient Orthogonality)** $\nabla_{\theta t_1} \ell(\mathbf{x}_i^{\text{clean}})^\top \nabla_{\theta t_2} \ell(\mathbf{x}_j^{\text{cont}}) = 0, \forall i \in [1, N_{\text{clean}}], \forall j \in [1, N_{\text{cont}}]$ and $\forall$ steps $t_1, t_2$.

We also make another minor simplifying assumption that the weight initialization is orthogonal to both these quantities.

**Assumption 2. (Gradient-Initialization Orthogonality)** $\nabla_{\theta_t}\ell(\mathbf{x}_i)^\top \theta_{init} = 0$, $\forall i$ and $\forall$ steps $t$.

Given these assumptions, we are ready to state our result.

**Proposition 3.** *(Forgetting Time) The number of optimization steps $T_{forget}$ in the forgetting stage, such that the weights $\theta_{T_{forget}}$ satisfy the $\epsilon$-forgetting criteria is given by: $T_{forget} \gtrsim \frac{\log(1/\epsilon)}{\lambda_{avg}\gamma}$, where $\lambda_{avg} = \frac{1}{T}\sum_{i=1}^{T}\lambda_i$ is the average learning rate of the optimizer.*

*Proof.* We first compute the forgetting ratio at $\theta = \theta'$, and as a consequence of Assumption 2, verify that the forgetting ratio is equal to one.

Let us now denote these weights as $\theta'_0 = \theta'$, used as initialization for the forgetting stage. Analyzing the first optimization step, and the subsequent forgetting ratio, we have:

$$\text{(Optimization Step)} \quad \theta'_1 = \theta'_0 - \lambda_0\nabla_{\theta_0}\ell(\mathbf{x}_0^{\text{clean}}) - \lambda_0\gamma\theta'_0$$

$$\text{(Forgetting ratio)} \quad \frac{|\theta'_1{}^\top\theta^{\text{cont}}|}{\|\theta^{\text{cont}}\|_2^2} = \frac{|(\theta'_0 - \lambda_0\nabla_{\theta_0}\ell(\mathbf{x}_0^{\text{clean}}) - \lambda_0\gamma\theta'_0)^\top\theta^{\text{cont}}|}{\|\theta^{\text{cont}}\|_2^2}$$

$$= \frac{|((\theta^{\text{init}} + \theta^{\text{cont}}) - \lambda_0\nabla_{\theta_0}\ell(\mathbf{x}_0^{\text{clean}}) - \lambda_0\gamma((\theta^{\text{init}} + \theta^{\text{cont}})))^\top\theta^{\text{cont}}|}{\|\theta^{\text{cont}}\|_2^2}$$

$$= (1 - \lambda_0\gamma) \quad \text{(From Assumptions 1 \& 2)}$$

We can similarly analyze the subsequent optimization steps to compute the forgetting ratio, which for some step $t + 1$ is:

$$\underbrace{\frac{|\theta'_{t+1}{}^\top\theta^{\text{cont}}|}{\|\theta^{\text{cont}}\|_2^2}}_{\text{Forgetting ratio at } t+1} = \frac{|(\theta'_t - \lambda_t\nabla_{\theta_t}\ell(\mathbf{x}_t^{\text{clean}}) - \lambda_t\gamma\theta'_t)^\top\theta^{\text{cont}}|}{\|\theta^{\text{cont}}\|_2^2}$$

$$= \underbrace{\frac{|\theta'_t{}^\top\theta^{\text{cont}}|}{\|\theta^{\text{cont}}\|_2^2}}_{\text{Forgetting ratio at } t} (1 - \lambda_t\gamma)$$

Unrolling the recurrence till step $T$, we have that:

$$\frac{|\theta'_T{}^\top\theta^{\text{cont}}|}{\|\theta^{\text{cont}}\|_2^2} = \underbrace{\frac{|\theta'_0{}^\top\theta^{\text{cont}}|}{\|\theta^{\text{cont}}\|_2^2}}_{= 1} \times \prod_{i=1}^{T}(1 - \lambda_i\gamma)$$

Assigning the forgetting ratio to be less than $\epsilon$ according to our criteria, we have:

$$\prod_{i=1}^{T}(1 - \lambda_i\gamma) \leq \epsilon$$

$$\sum_{i=1}^{T}\log(1 - \lambda_i\gamma) \leq \log\epsilon$$

$$\sum_{i=1}^{T}(-\lambda_i\gamma) \lesssim \log\epsilon \quad (\log(1 - x) \approx -x \text{ for small } x)$$

$$T \times \left(\frac{1}{T}\sum_{i=1}^{T}\lambda_i\right) \times \gamma \gtrsim \log\frac{1}{\epsilon}$$

Re-arranging this equation gives us the desired result, where $\lambda_{avg} = (\frac{1}{T}\sum_i\lambda_i)$. $\qquad\square$

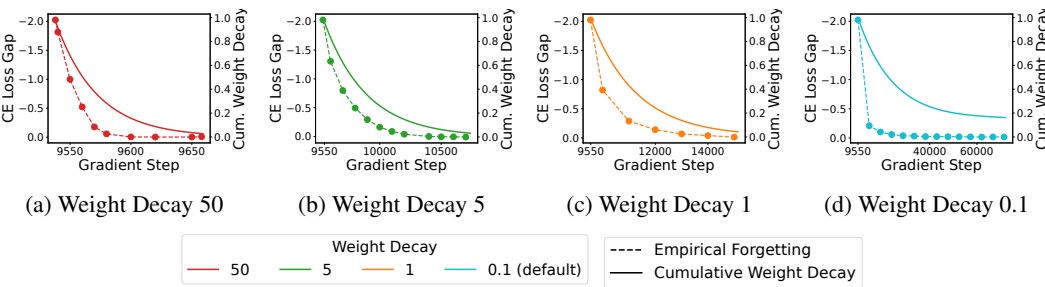

(a) Weight Decay 50    (b) Weight Decay 5    (c) Weight Decay 1    (d) Weight Decay 0.1

Figure 7: **Increasing the weight decay parameter leads to faster forgetting described by the cumulative weight decay.** We continue training the 124M parameter model from Section 4.2 with different choices of the weight decay parameter. The figure depicts empirical forgetting (dashed line) and our theoretical forgetting bound, the cumulative weight decay (solid line). As we increase the weight decay parameter, the theoretical bound predicts much faster forgetting. The figure shows that empirical forgetting is indeed as fast as predicted by the bounds. Note that the figures have very different **x-axis scales**, ranging from 120 gradient steps in (a) to 62500 gradient steps in (d). This Figure depicts the same quantities as Figure 6 in the main paper.

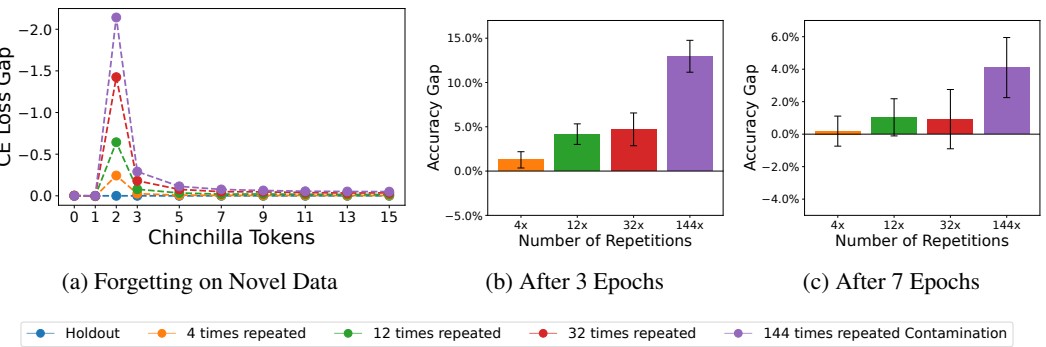

(a) Forgetting on Novel Data    (b) After 3 Epochs    (c) After 7 Epochs

Figure 8: **Forgetting without weight decay.** We perform the forgetting experiment from Section 4.2 in the main paper without weight decay (instead of the default weight decay of 0.1). This Figure depicts the same quantities as figure 3 in the main paper. We see that there is significant forgetting without weight decay. However, compared with a weight decay of 0.1, the rate of forgetting is somewhat slower, especially for 144 times repeated contamination. In particular, the accuracy gaps in (b) are slightly larger, and the difference in (c) is still statistically significant (compare Figure 3).

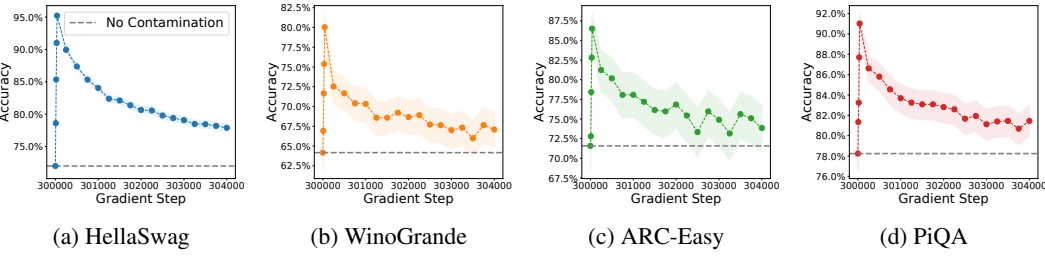

(a) HellaSwag    (b) WinoGrande    (c) ARC-Easy    (d) PiQA

Figure 9: **Scaling up with a 7B parameter model.** We perform the forgetting experiment from Section 4.3 in the main paper with OLMo-7B. We contaminate the OLMo-7B checkpoint at gradient step 300,000 four times with different benchmarks. This causes an average accuracy increase of 17 percentage points. We then continue pre-training for 1% of the remaining training time, leading to a reduction of 77% of the accuracy increase due to contamination. Mean and bootstrapped 90% confidence intervals. **We will continue training this model for the final version of the paper.**

