# OpenReview forum: "How much can we Forget about Data Contamination?"
_ICLR.cc/2025/Conference — Submitted to ICLR 2025_

### Official Review · Reviewer_vqnY · 2024-11-04

**Soundness:** 4
**Presentation:** 3
**Contribution:** 3
**Rating:** 8
**Confidence:** 4

**Summary:**

The paper investigate the effects of data contamination in LLMs and explore whether these models can forget contaminated data during training. The author challenge the common assumption that minor contamination renders benchmark evaluations invalid, by presenting novel experimental and theoretical insights. They show that the benchmark overfitting scaling is monotone in the number of model params, the number of of repetitions of examples in training data, and the number of training tokens.  Authors also develop a theoretical framework to understand how cumulative weight decay in AdamW facilitates forgetting of contained data, suggesting that under certain conditions, this decay can render contaminated samples negligible by the end of training. TLDR; Contamination concerns may be overestimated for extensive training runs of modern LLMs, and weight decay play a critical role in “forgetting” in data-intensive training regimes.

**Strengths:**

This paper explores the critical topic of data contamination in LLM training, as model scale increases and samples contamination gets more likely. Authors offer a timely into how the contamination affects model evaluations, challenging the prevailing assumption that any contamination renders the evals invalud. The paper has a very scientific approach and combines nice theoretical insights with extensive experimental validation, presenting contamination as a challenge that might be mitigated by the large-scale and data-intensive nature itself of model LLM training setups. They propose that the cumulative weight decay in AdamW (often used to train modern LLMs) acts as a natural forgetting mechanism for contaminated data. The overall paper is very sound, balancing a solid theoretical framework, with carefully controlled experiments across various model scales, token volumes and contamination frequencies. The combinations of gpt-3 and large OLMo-1B model along with clearly documented, reproducible methods, code and datasets, adds significant credibility to the findings.

**Weaknesses:**

The theoretical framework for forgetting seems to rely on the assumption that contamination occurs uniformly and that the data scale follows the Chinchilla scaling law. This limits generalizability, as many real-world LLms operate outside this scaling regime or include more intricate forms of (more complex) contamination. The theoretical derivations in the cumulative weight decay analysis, although sound, assume orthogonality in gradient updates without acknowledging that in real-world training, gradient alignment might cause contaminated examples to persist longer (dependencies between gradients could affect forgetting).

Minor things:
Line 055-056: I would change ordering of (2) and (3) to be consistent with lines 059-062.
Line 195: "We being" -> "We begin"

**Questions:**

— Given the focus on cumulative weight decay, how does your theory account for rare or privacy-sensitive data points that might resist weight decay due to repetition? Do you have experimental results that assess forgetting for low-frequency, high-sensitivity samples?
— As gradient updates are often correlated, especially, in high-frequency or structure contamination; have you considered conducting experiments simulating the effect of gradient alignment impact on the the cumulative weight decay effect?
— Do you have insights on why the empirical forgetting occurred faster than the predictions by your theory? What are the factors contributing to the discrepancy?

---

> ### Author Response · Authors · 2024-11-20
> **Author Response to Reviewer vqnY**
>
> We thank the reviewer for appreciating our paper, especially concerning our scientific approach.
>
> **“The theoretical framework for forgetting seems to rely on the assumption that contamination occurs uniformly and that the data scale follows the Chinchilla scaling law.”**
>
> The reviewer is correct to observe that our theoretical framework relies on simplifying assumptions. In particular, we only model forgetting of samples contained in individual gradient updates. The case of repetition could be handled insofar as repeated gradient updates could all be forgotten, but this might not be very realistic.
>
> The theory does not assume that the data follows the Chinchilla scaling law.
>
> Please also see the global comment on the theoretical framework.
>
> **“Given the focus on cumulative weight decay, how does your theory account for rare or privacy-sensitive data points that might resist weight decay due to repetition? [...]  Do you have insights on why the empirical forgetting occurred faster than the predictions by your theory? What are the factors contributing to the discrepancy?”**
>
> Great question. Indeed, forgetting might behave differently for rare and privacy-sensitive data points. Most importantly, one would likely want to use a different metric to assess forgetting in this setup than the average accuracy gap (see, for example, Jagielski (2023)).
>
> Our theory does apply to the privacy setup as long as a data point is seen only during a single gradient update. If repetition is introduced, the question becomes more empirical, and we agree with the reviewer that it would be very interesting to study this experimentally. We also agree that it would be interesting to consider the factors mentioned by the reviewer, especially how gradient alignment interacts with fast forgetting. We will leave the investigation of these questions for future work.
>
> We would be happy to answer any additional questions during the discussion phase.

---

### Official Review · Reviewer_3n24 · 2024-11-04

**Soundness:** 3
**Presentation:** 3
**Contribution:** 2
**Rating:** 5
**Confidence:** 3

**Summary:**

The paper explores specific learning dynamics in language models, focusing on how these models gradually "forget" small-scale contaminations. This direction is crucial for understanding the reliability of benchmark evolution and its connection to potential contamination. To investigate this, training is conducted on data infused with samples from different benchmarks at varying contamination levels. Additionally, models of different scales are analyzed to assess the impact of model size on this phenomenon.

**Strengths:**

The paper empirically studies various settings:
- How various levels of contamination from various sets influences the performance in those respective sets?  Figure 2
- How long does it take to forget these contaminations? Fig. 3a What kind of data makes it forget? Fig 3e and their position in training ? Fig 3b,c
- How is the effect the scales of model, data and frequency of contaminated examples on forgetting the contamination ? Figure 2
These questions are  important and the paper tries to address them.

**Weaknesses:**

- The theoretical analysis is very limiting - the only case considered is forgetting the exact gradient of the contaminated example. The analysis is preliminary as the paper already mentioned that it does not take into account the forgetting of the gradient by cancellation. The gradient of related contaminated samples late in the training are also not accounted.
- Experiments with larger models are needed to validate the findings across scale.

**Questions:**

- Why the frequency of the repetitions are chosen as 4, 12, 32, 144 (instead of say 4,16,32,128 - the powers of 2) ?
- The forgetting phenomenon is attributed to the weight decay, does it mean training without weight decay does not forget contamination.

---

> ### Author Response · Authors · 2024-11-20
> **Author Response to Reviewer 3n24**
>
> We thank the reviewer for taking the time to review our paper.
>
> **“The theoretical analysis is very limiting - the only case considered is forgetting the exact gradient of the contaminated example.”**
>
> While the analysis is simple, it suffices to demonstrate significant forgetting in large-scale LLM training runs. Please look at our global response for more details on the theoretical analysis in Section 5.
>
>
> **“Experiments with larger models are needed to validate the findings across scale”**
>
> To address this, we conduct a forgetting experiment with the OMLo-7B model. This experiment requires substantial computational resources and is still running. Supplement Figure 9 depicts preliminary results. The results are qualitatively similar to those of the OLMo-1B model (Figure 4 in the main paper), indicating that our results extend towards the 7B parameters scale.
>
>
> **“Why the frequency of the repetitions are chosen as 4, 12, 32, 144 (instead of say 4,16,32,128 - the powers of 2) ?”**
>
> That's a fair point. The frequencies were chosen based on preliminary experiments with the 124M model to cover the range from statistically significant contamination to complete overfitting. In hindsight, seeing the full experimental results (Figure 2), we agree that it would be sensible to choose the powers of 2.
>
>
> **“The forgetting phenomenon is attributed to the weight decay, does it mean training without weight decay does not forget contamination.”**
>
> Please see the global comment: It turns out that a model trained without weight decay also forgets contamination, but not necessarily at the same rate.
>
> We would be happy to answer any additional questions during the discussion phase.

---

> ### Comment · Reviewer_3n24 · 2024-11-25
> **Reply to the authors**
>
> I thank the authors for their reply to my questions.
>
> Given that the forgetting phenomenon is not attributed to weight decay, i.e., the network forgets even while training without weight decay. It seems like there is some hidden phenomenon that causes this forgetting, weight decay merely accelerates it but it is not actual cause. Hence, it is a bit misleading to attribute this to weight decay and I hope the authors really tone it down.
>
> I do not agree with the comment that *forgetting past gradients is sufficient for forgetting the data*, consider this simple setup of GD, after t time steps, the iterate writes as
> $$  \theta_{T} = w_ {0}^{T} \theta_{0} - \sum_{j=1}^{T-1}  w_{j}^{T} g_{t}, $$
> note that here $g_{t} = \nabla L(\theta_{t})$, so that gradient at some $ i < t$ also contributes to this gradient, even if it explicitly forgot $g_i$, its influence can be propagated through these gradient computations.

---

> > ### Author Response · Authors · 2024-11-27
> > **Reply to Reviewer 3n24**
> >
> > Thanks for your additional comments. They will help us to improve the paper.
> >
> > It was never our claim that the forgetting phenomenon is attributed to weight decay. For example, we write that *"the decay of past gradients described in Proposition 1 is only one effect that contributes to forgetting. "* (L431).
> >
> > The main goal of Section 5 is not to explain Figure 3. Instead, we want to identify weight decay as an important mechanism contributing to forgetting and use it to gauge the degree of forgetting in larger training runs.
> >
> > We agree that this can be clarified commit to clarifying it in the final version.

---

### Official Review · Reviewer_c415 · 2024-11-04

**Soundness:** 2
**Presentation:** 3
**Contribution:** 3
**Rating:** 6
**Confidence:** 2

**Summary:**

This paper investigates the effects of small-scale data contamination in training language models. The experiments involve randomly injecting benchmark evaluation data into the training set, and then assessing the impact on loss and accuracy when evaluated against the "leaked" benchmark. The study examines how this effect changes with model size, the repetition frequency of contaminated examples, and scaling in accordance with the Chinchilla scaling law. The authors observe that generally a small amount of data contamination is forgotten after a few steps, and suggested a possible theory that weight decay influences this forgetting phenomenon. They provide a theoretical bound for the rate of forgetting due to weight decay, though they find that forgetting occurs much faster in practice.

**Strengths:**

- The research question is compelling and relevant, as data contamination is an unavoidable issue in practical LLM training.

- The experiments cover a wide range of model sizes and contamination scales.

**Weaknesses:**

- My main concern is about the theory that weight decay influences forgetting. It's unclear to me why "forgetting past gradients" would equate to forgetting past training data since the model weights have already been updated based on that past data. In other words, past gradients have already shaped the optimization trajectory, so even if they no longer impact future updates, their influence isn't necessarily "forgotten." The fact that the theoretical bound is quite loose in practice also suggests that weight decay might not be the main factor behind forgetting.

- Evaluating LLMs is highly complex, and I'm unsure if simply comparing performance on the leaked benchmark is sufficient to fully capture whether the models have been influenced by data leakage or if it has been forgotten.

- Although the observation is interesting, it’s unclear how this information can be practically applied, especially given the difficulty of accurately measuring data leakage in real-world LLM training. What is the practical impact of these findings?

- More of a future direction but would be interesting to see how data contamination affects other types of tasks and architectures, such as vision models and non-transformer models, as well as scenarios where training data are encountered multiple times. It would also be valuable to explore whether the weight decay explanation holds in these cases.

**Questions:**

- Intuitively, at least in more traditional ML, weight decay constraints drastic changes in model weights, which can help prevent both overfitting to earlier data and excessive drift from initial representations when new data is introduced. To me it makes more sense that weight decay plays a balancing role, encouraging the model to learn simpler representations. Indeed it can be challenging to determine how this applies to LLM training where factors like single-pass data and other complexities come into play. If the observed influence of weight decay on forgetting is specific to LLM training, what aspects of LLM training might be driving this effect?

- In the related work, the paper mentions that it focuses on "natural forgetting," which differs from catastrophic forgetting in continual learning. Could the authors provide a more quantitative explanation of this difference, beyond the distinction of being desirable or undesirable based on context?

- In the last sentence of the paragraph next to Table 1, it is mentioned that "under Chinchilla training, a single instance of contamination can lead to overfitting of up to 3 percentage points." I'm curious where the number "3%" comes from, as the scaling of the accuracy gap doesn’t appear linear with the number of repetitions. Is this a rough estimate? Additionally, why isn’t the effect of a single contamination instance directly shown in the experiments?

- In Figure 3 the description for plot (d) is very confusing since it's not actually the same as (a), please specify it more clearly. I'm interested in the difference between multi-pass data and single-pass data, but in this experiment, the number of tokens models are trained on is not the same between the two settings. So is the forgetting impacted by the fact that the training set is seen multiple times, or purely because the number of different tokens is lower?

---

> ### Author Response · Authors · 2024-11-20
> **Author Response to Reviewer c415 (Part 1/2)**
>
> We thank the reviewer for taking the time to review our paper.
>
> **“My main concern is about the theory that weight decay influences forgetting. It's unclear to me why "forgetting past gradients" would equate to forgetting past training data since the model weights have already been updated based on that past data.”**
>
> General remarks on the theory in Section 5 are provided in the global comment.
>
> As we also outline in the global comment, we claim that "forgetting past gradients" is sufficient for "forgetting past data," not that it is equivalent. We agree that this assumption underlies the theory in Section 5. The additional experiment depicted in Figure 7 of the revised Supplement provides strong evidence supporting this assumption.
>
> The reviewer makes an interesting observation about the optimization trajectory and how this is related to example forgetting. The novel experiment depicted in Supplement Figure 7 shows that the effect mentioned by the reviewer is unlikely to exist, at least in our setting.
>
> We agree with the reviewer that weight decay might not always be the main factor behind forgetting (see the global comment). However, it suffices to demonstrate significant forgetting in large-scale LLM training runs.
>
> **“I'm unsure if simply comparing performance on the leaked benchmark is sufficient to fully capture whether the models have been influenced by data leakage”**
>
> For this paper, we decided to use benchmark accuracy as our primary metric because this is the performance measure that LLM developers are expected to report.
>
>
> **“What is the practical impact of these findings?”**
>
> Our empirical results show that individual samples can have a negligible impact on model performance and that there is a significant forgetting effect during training. This has many practical implications, for example:
>
> *Memorization-Aware Training Pipelines.* Our insights regarding forgetting can be used to design training pipelines that mitigate benchmark contamination. Concretely, our work suggests that contaminated models may continue pre-training on “clean” data to ensure that contamination is forgotten.
>
> *Data Attribution.* Traditionally, data attribution methods attribute model performance to individual data points. We show that this might not be a valid approach in large-scale settings, implying that model performance needs to be attributed to larger clusters of data instead (for example, as in Ley et al. “Generalized Group Data Attribution”, 2024, https://arxiv.org/abs/2410.09940).
>
> *Data Pre-Processing.* The fact that individual samples are negligible has implications for data pre-processing, filtering, and cleaning (an increasingly significant topic of research for LLMs).
>
> **“More of a future direction but would be interesting to see how data contamination affects other types of tasks and architectures, such as vision models and non-transformer models, as well as scenarios where training data are encountered multiple times. It would also be valuable to explore whether the weight decay explanation holds in these cases.”**
>
> We agree that this would be interesting to explore in future works!
>
> **“If the observed influence of weight decay on forgetting is specific to LLM training, what aspects of LLM training might be driving this effect?”**
>
> Training for a single epoch and more than 100,000 gradient steps.
>
> (Part 1/2)

---

> > ### Author Response · Authors · 2024-11-20
> > **Author Response to Reviewer c415 (Part 2/2)**
> >
> > **Quantitative difference between natural and catastrophic forgetting**
> >
> > Catastrophic forgetting describes the phenomenon where the model forgets a task in a fine-tuning or continual learning setting that involves a distribution shift. In contrast, we study the phenomenon where individual data points are forgotten over the course of a single pre-training run.
> >
> > Conceptually, the most important difference between the two might be that catastrophic forgetting is defined at the level of probability distributions over tasks, whereas we study what happens to individual data points. However, it is plausible that “catastrophic” and “natural” forgetting are different edge cases of closely related phenomena during neural network optimization.
> >
> > As a side note, we decided to use the term “natural” in this paper because forgetting individual data points is a natural and potentially desirable effect during learning. Indeed, a machine learning algorithm that generalises should be allowed to “throw away” individual data points (Bousquet and André Elisseeff, 2002).
> >
> >
> > **"under Chinchilla training, a single instance of contamination can lead to overfitting of up to 3 percentage points."**
> >
> > This is a rough estimate. We will note this in the paper.
> >
> >
> > **“I'm interested in the difference between multi-pass data and single-pass data, but in this experiment, the number of tokens models are trained on is not the same between the two settings. So is the forgetting impacted by the fact that the training set is seen multiple times, or purely because the number of different tokens is lower?”**
> >
> > It is impacted by the fact that the training set is seen multiple times.
> >
> > Consider the amount of forgetting at 7 epochs on the x-axis both in Figure 3(a) and Figure 3(d) (the "7" on the x-axis in both plots). We see that the cross-entropy loss differences for the model trained on novel data are within the confidence interval of the holdout (Figure 3(a)), whereas the same differences are still quite significant for the model that was repeatedly trained on 100M tokens (Figure 3(d)).
> >
> > The overall number of tokens that the model in Figure 3(d) is trained on is less because the model started to diverge after being trained on the same training set too many times. Concretely, the training set consists of 100M tokens, a single Chinchilla epoch on the x-axis of 2.5B tokens, so the model depicted in (d) was trained 25 * 6 = 150 times on the same training set after the second epoch. Up to the point where the model in Figure 3(d) started to diverge the training for the two models was exactly the same (except for the repeated training data, of course).
> >
> > We again thank the reviewer for taking the time to review our paper. We would be happy to answer any additional questions during the discussion phase.

---

> > > ### Comment · Reviewer_c415 · 2024-11-21
> > > **Reviewer response**
> > >
> > > Thank you for the detailed clarification. I have increased my score.

---

### Official Review · Reviewer_xbPV · 2024-11-08

**Soundness:** 3
**Presentation:** 4
**Contribution:** 3
**Rating:** 8
**Confidence:** 2

**Summary:**

In the paper, the authors study the problem of data contamination in large language models, and the ability of these models to forget past contaminated samples during training. In the first part of the paper, the authors provide extensive experiments on the effects of data contamination and forgetting in a controlled setting, where some of the testing samples are included in the training set in random positions. In the second part, the authors derive a theoretical upper bound on the rate of forgetting in LLMs trained with AdamW as a function of the learning rate and the weight decay parameter.

**Strengths:**

Unfortunately I am not very familiar with the topic, so I cannot fully judge the novelty and impact of the work. Tot he best of my understanding, I find this paper very well written and the topic very interesting and timely. I also find the experiments provided by the authors are extensive and satisfying.

**Weaknesses:**

I would guess that the limit of the forgetting approach presented in the paper, in particular in Sections 4.2 and 4.3, is that no matter how long you train a model, there will reasonably always be contaminated samples in the last part of the training, say, in the final Chinchilla samples, that the final model has no time to forget. If you agree with this and I am not missing something, could you please elaborate on this? And can the results that you presented in the paper address this point?

**Questions:**

Apart from the question in the Weaknesses section, I would also be interested to know if there is any correlation between the accuracy gap metric presented in Section 4.1 and the percentage of training samples that are contaminated. For example, it is expected that each line in Figure 2(b) decreases with the number of tokens, since the percentage of contaminated samples decreases as well. I was wondering if there is a proportionality law between the percentage of contaminated samples and the observed accuracy gap.

---

> ### Author Response · Authors · 2024-11-20
> **Author Response to Reviewer xbPV**
>
> We thank the reviewer for the review and appreciating our paper.
>
> **“there will reasonably always be contaminated samples in the last part of the training, say, in the final Chinchilla samples, that the final model has no time to forget. [...] could you please elaborate on this”**
>
> Great observation. While more experimental evidence would be required, we agree with the reviewer that our work hints at the fact that the model might always remember specific samples. In this regard, there are a couple of points to keep in mind.
>
> One point is the learning rate schedule. Because the learning rate usually decays towards the end of training, it is not necessarily the samples that are seen last that are “best remembered” by the model. For example, Figure 3(e) and Figure 3(f) in the main paper show that benchmark questions that are uniformly distributed throughout training exhibit more overfitting than benchmark questions seen toward the end.
>
> Another point is that modern language models go through different stages of training. For example, an LLM might be pre-trained on data from the internet, continue training on synthetically generated data, and finally be fined-tuned as a chat model. In such a training process, the stage where the leakage of benchmark questions into the training data is likely to occur is the first stage of pre-training on data from the internet. For this paper, we are primarily interested in the leakage of benchmark questions, and our experiments are mainly meant to cover the initial pre-training stage. This means that the final samples seen by our models would not necessarily be the final samples seen by the model in a realistic training setup.
>
> Regarding our benchmark questions setup, it does not matter much if the model remembers specific individual questions as long as the overall fraction remains negligible (compared to, say, 10,000 test questions). However, studying this question would be very relevant in a privacy setup.
>
> **“I was wondering if there is a proportionality law between the percentage of contaminated samples and the observed accuracy gap.”**
>
> We agree that holding everything else constant, the accuracy gap decreases with the percentage of contaminated samples (though the relationship might be non-linear). At the same time, Figure 2(a) shows that the accuracy gap increases in the number of model parameters even if the percentage of contaminated samples is held constant. So the percentage of contaminated samples is an important ratio, but it does not uniquely determine the observed accuracy gap.
>
> We would be happy to answer any additional questions during the discussion phase.

---

> ### Comment · Reviewer_xbPV · 2024-11-23
>
> I thank the authors for their responses. I will keep my score and reaffirm my appreciation for the paper.

---

### Author Response · Authors · 2024-11-20
**Author Response to all Reviewers**

We thank the reviewers for taking the time to review our paper. We appreciate that they find our research *interesting* and *timely*.

We answer each reviewer's individual questions below. Here, we would like to clarify how weight decay is related to forgetting, a point that came up in several reviews. This concerns Section 5 of the paper. Of course, we will also clarify these points in the final version of the paper.

The main insight in Section 5 is that the weight decay and learning rate schedule can be used to provide a rough estimate of the rate of example forgetting. In particular, we claim that empirical forgetting occurs *at least as fast* as the cumulative weight decay. However, weight decay is not the only mechanism for forgetting. There might be many other mechanisms related to the optimization dynamics of neural network training (Reviewer c415 and Reviewer vqnY). These are not covered by our simple theory.

To clarify the role of weight decay in forgetting, we conducted two additional experiments. The results of the new experiments are depicted on the final page of the revised Supplement.

**Experiment 1: Weight decay is sufficient for forgetting.** We demonstrate that the cumulative weight decay does indeed bound empirical forgetting. We consider the 124M model from Section 4.2 at the point of the strongest contamination (that is, after 2 epochs). We then intervene on the weight decay parameter, setting it to artificially large values (1, 5, and 50). According to our estimates, this should significantly speed up forgetting, up to the point where all contamination is forgotten after only 120 gradient steps (for weight decay 50). Supplement Figure 7 shows that this is indeed the case. In particular, the empirical rate of forgetting is always faster than the cumulative weight decay (Reviewer c415).

**Experiment 2: Weight decay is not necessary for forgetting.** We train the 124M model from Section 4.2 without weight decay. The result of this experiment is depicted in Supplement Figure 8. The model trained without weight decay exhibits significant forgetting – albeit at a slightly slower rate than the model trained with the default weight decay of 0.1.  This means that weight decay is not necessary for forgetting (Reviewer 3n24).

**Weight decay is relevant for forgetting in LLMs because of the large-scale training.** Given typical choices for the learning rate (e.g. 2e-4) and weight decay parameter (typically 0.1 or 0.01), the process of example forgetting via cumulative weight decay is relatively slow. In particular, we can see from Figure 5(b) and Figure 5(c) in the paper that approximately 100,000 gradient steps are required until a past gradient update has fully decayed to zero. The training regime of training for a single epoch and more than 100,000 gradient steps has only recently emerged with LLMs, which is why we believe that the process of forgetting via weight decay has yet to attract attention (Reviewer c415).

We again thank the reviewers for taking the time to review our paper.

---

### Meta-Review · Area_Chair_Du9g · 2024-12-19

**Metareview:**

The paper explores data contamination in large language models and investigates whether contaminated data is "forgotten" during training. The work combines extensive experiments with a theoretical analysis of forgetting, particularly focusing on the role of cumulative weight decay in AdamW.

Strengths:

- The paper tackles a timely and relevant problem, providing insights into data contamination and its implications for benchmark evaluations.
- The experiments are thorough, exploring contamination across model scales, token volumes, and repetition frequencies.

Weaknesses:

- The theoretical analysis is limited, as it assumes uniform contamination and orthogonality in gradient updates, oversimplifying real-world dynamics.
- The claim that weight decay drives forgetting remains unconvincing. Reviewers were concerned that forgetting gradients is insufficient to conclude forgetting contaminated data since gradients influence subsequent updates.
- Empirical results, while thorough, leave questions about the practical impact of the findings and their generalizability beyond the controlled setup.

After extensive discussions, the consensus is that while the problem is interesting and experiments are solid, the theoretical contributions are limited, and the role of weight decay remains unclear. I recommend rejection at this time.

**Additional Comments On Reviewer Discussion:**

During the discussion, reviewers raised two primary concerns: (1) the role of weight decay in forgetting contaminated data, and (2) the practical significance of the empirical findings regarding data forgetting. Some reviewers questioned whether weight decay sufficiently explains forgetting, as past gradients influence the optimization trajectory. The paper was thoroughly discussed between the reviewers and the Area Chair. Although initial review scores leaned towards acceptance, a consensus was reached to reject the paper due to unresolved concerns about the theoretical claims.

---

### Decision · Program_Chairs · 2025-01-22

Reject